# African Swine Fever Outbreak Investigations—The Significance of Disease-Related Anecdotal Information Coming from Laypersons

**DOI:** 10.3390/pathogens11060702

**Published:** 2022-06-17

**Authors:** Kristīne Lamberga, Felix Ardelean, Sandra Blome, Paulius Busauskas, Boban Djuric, Anja Globig, Vittorio Guberti, Aleksandra Miteva, Edvins Oļševskis, Mārtiņš Seržants, Arvo Viltrop, Laura Zani, Anna Zdravkova, Klaus Depner

**Affiliations:** 1Food and Veterinary Service, LV-1050 Riga, Latvia; kristine.lamberga@pvd.gov.lv (K.L.); edvins.olsevskis@pvd.gov.lv (E.O.); martins.serzants@pvd.gov.lv (M.S.); 2Institute of Food Safety, Animal Health and Environment “BIOR”, LV-1076 Riga, Latvia; 3Faculty of Veterinary Medicine, Latvia University of Life Sciences and Technologies, LV-3001 Jelgava, Latvia; 4County Sanitary Veterinary Health and Food Safety Directorate, 4400067 Satu Mare, Romania; ardelean.felix-sm@ansvsa.ro; 5Friedrich-Loeffler-Institute, Greifswald-Insel Riems, 17493 Greifswald, Germany; sandra.blome@fli.de (S.B.); anja.globig@fli.de (A.G.); laurazani@outlook.de (L.Z.); 6Emergency Response Division, State Food and Veterinary Service, LT-07170 Vilnius, Lithuania; paulius.busauskas@vmvt.lt; 7Veterinary Directorate, Ministry of Agriculture, Forestry and Water Management, 1 Omladinskih Brigade St., 11000 Belgrade, Serbia; boban.djuric@minpolj.gov.rs; 8National Institute for Environmental Protection and Research, 40064 Ozzano dell’Emilia, Italy; vittorio.guberti@isprambiente.it; 9Animal Health Department Bulgarian Food Safety Agency, 1606 Sofia, Bulgaria; a_miteva@bfsa.bg (A.M.); a_zdravkova@bfsa.bg (A.Z.); 10Veterinaarmeditsiini Ja Loomakasvatuse Instituut, Estonian University of Life Sciences, 5106 Tartu, Estonia; arvo.viltrop@emu.ee

**Keywords:** ASF, outbreak investigation, anamneses, subjectivity, laypersons

## Abstract

Veterinarians who have conducted numerous investigations of African swine fever outbreaks in pig farms in various European countries over the years shared their experiences during a workshop in Germany in early 2020. One focus was on the so-called “anecdotal information” obtained from farmers, farm workers or other lay people during the outbreak investigations. Discussions revolved around how to correctly interpret and classify such information and how the subjective character of the statements can influence follow-up examinations. The statements of the lay persons were grouped into three categories according to their plausibility: (i) statements that were plausible and prompted further investigation, (ii) statements that were not plausible and could therefore be ignored, and (iii) statements that were rather implausible but should not be ignored completely. The easiest to deal with were statements that could be classified without doubt as important and very plausible and statements that were not plausible at all. Particularly difficult to assess were statements that had a certain plausibility and could not be immediately dismissed out of hand. We aim to show that during outbreak investigations, one is confronted with human subjective stories that are difficult to interpret but still important to understand the overall picture. Here, we present and briefly discuss an arbitrary selection of reports made by lay persons during outbreak investigations.

## 1. Introduction

The current African swine fever (ASF) epidemic in Europe started in Georgia when a genotype II African swine fever virus (ASFV) was detected in 2007. From there, it spread to mainland Europe where it affected the wild boar population as well as the domestic pig sector. Thousands of outbreaks have been reported in small farms, mainly in Central and Eastern Europe, but also in huge commercial farms with thousands of pigs [1,2,3,4,5]. 

An epidemiological investigation is mandatory in the case of ASF and other listed diseases according to EU legislation [6,7]. In the event of an ASF outbreak in a domestic pig farm, a timely and thorough epidemiological investigation has to be carried out by the competent authority to find out the likely origin of the virus, to estimate the time that the disease has been on the holding prior to notification and to obtain information on the potential spread of the virus to other holdings. To clarify all these crucial aspects, it is extremely important to reconstruct the course of the disease in the herd. This process of working up the case and searching for the reasons how and when the pathogen entered and/or left the farm usually starts with an anamnesis. Farm owners, farm workers, farm veterinarians, business operators and other people familiar with the events on the farm are asked to give their opinions and insights into the case. In most cases, these are lay persons who have no particular epidemiological knowledge about ASF. Nevertheless, they provide valuable information that can help to clarify the circumstances that led to the outbreak.

This short communication is based on an ASF workshop in Germany in early 2020, summarizing the experiences of veterinarians, mainly epidemiologists and disease managers, who have conducted outbreak investigations in several European countries during the past years and thereby gathered valuable information and experience. One focus of the workshop was on the so-called “anecdotal information” obtained from laymen during the outbreak investigations, particularly during anamnesis. In our context, we defined anecdotal information as observations or views that were directly linked to an ASF outbreak by the reporters, but were not based on scientific evidence according to the investigators. In particular, the subjective nature of such reports and their possible influence on further investigations was discussed.

We present and briefly discuss an arbitrary selection of statements collected during outbreak investigations in Central and Eastern Europe over the last six years.

## 2. Material and Methods

### 2.1. Statements and Opinions from Laypersons

During the ASF outbreak investigations conducted in Central and Eastern European countries and Balkans (Estonia, Latvia, Lithuania, Romania, Bulgaria and Serbia) since 2014, a wide range of opinions have been obtained from lay persons. These are the personal views of the affected farmers, farm workers and other lay persons regarding how and when the pathogen entered the farm and how the disease spread and manifested. They contain the interviewee’s personal subjective views on the case. 

Most of the outbreaks occurred in countries and regions where local wild boar populations were also affected by ASF and served as virus reservoirs for the domestic pig outbreaks [8]. 

### 2.2. Grouping of Statements

The statements and opinions were grouped into three categories according to their plausibility regarding the course of ASF:

Statements that are plausible and need to be followed up by further investigations;

Statements that are not plausible and can therefore be ignored; 

Statements that are rather implausible but should not be ignored completely.

The plausibility classification was made based on generally accepted knowledge on ASF as described in the current textbooks, publications and documents of the international organizations and institutions (OIE, FAO, EFSA and EU Commission) as well as the scientific literature on ASF. This includes knowledge about the route of transmission, contagiousness, susceptibility, incubation period, clinical course, convalescence, diagnosis and epidemiology. 

## 3. Results

The arbitrarily selected statements and opinions as well as the grouping according to plausibility are shown in Table 1.

## 4. Discussion

Epidemiological outbreak investigations are mostly carried out by a team of experts and usually start with the anamnesis. The collection of the case history is followed by the physical inspection of the farm with special attention to the farm’s biosecurity and the clinical examination of the animals in the different farm units (stables). In the course of these investigations, findings accumulate, and initial speculations are confirmed or discarded. A repeated critical analysis after each step of the examination with meaningful inclusion of the anamnestic data can lead to a preliminary hypothesis about the course of ASF on the farm. Thus, it should be noted that the reconstruction of the infection course is initiated and supported by the anamnestic findings, which might be a mix of objective and subjective information. Much of this information is obtained from people outside the field who have no expertise or experience with the disease. In most cases, these are the owners themselves or employees or other lay people. Often, statements are anecdotal information and not based on biological, proven evidence. In the worst case, such information can become a source of fake news [9]. Therefore, investigators should always be aware that some of the information they receive is subjective and sometimes misleading. Nor can it be ruled out that reports will always be truthful. Reporters could attempt to exonerate themselves from blame if certain requirements (e.g., concerning farm biosecurity) were not followed. This requires that the interviewer has the necessary disease-related knowledge and expertise to interpret and classify correctly the information received. The interviewer has to be very familiar with the clinical course and epidemiology of the disease.

The easiest to cope with are statements that can be classified without doubt as important and very plausible and statements that are not plausible at all. Statements that have a certain plausibility and cannot be immediately dismissed are particularly difficult to assess.

ASF is an infectious disease that does not spread via droplets or air. However, it has been shown that over very short distances, transmission can also take place via the air [10]. A susceptible animal becomes infected mainly parenterally (e.g., via a tick bite or infected needle) or oronasally; other transmission routes become implausible [11,12]. Other cases need a deeper reflection. For example, statement Nr. 5 (Table 1) which is reported in a similar way more frequently would require that the grandfather would have stepped in wild boar droppings or carcasses contaminated with ASF virus in the forest and this contaminated shoe would then have to get into the pigsty where pigs chew on it. Such a scenario cannot be completely ruled out, of course, but it is very unlikely. Yet, following such a statement, there is a danger of losing sight of other transmission possibilities, which are much more likely, and thus being steered onto a wrong track. On the other hand, in a Lithuanian case control study “activities in the forest” were positively associated with ASF outbreaks in the affected pig herds [13]. 

Statements that could possibly be plausible must be given special attention, because here there is the greatest danger of being misled, i.e., example 7. It was assumed that the disease was introduced by rats in the course of rodent control measures on the neighboring farm. Here, there is both a spatial and temporal connection to the ASF-positive farm where the rodent control took place. However, it is neither known nor proven that rats can transmit ASF. Nevertheless, this finding should lead to further investigation of other potential transmission possibilities from the neighboring farm.

We do not want to exclude the possibility that the statements interpreted and classified by us might have been assessed differently by other experts. Rather, we want to highlight the subjective character and also point out that it is relatively easy to be misled and possibly steer further investigations in the wrong directions. However, the more comprehensive and up to date the interviewer’s expertise on ASF is, the more likely it is that subjective information will be correctly understood and interpreted. The examples we have listed, which have crystallized from extensive experience over many years, are intended to provide the reader with an incentive for their own interpretation and reflection. Furthermore, we aim to show that during outbreak investigations, one is confronted with subjective perceptions that are not in line with the purely scientific disease-related logic.

Taking the outbreak history aims at understanding the problem to be dealt with and guiding the farm inspection and clinical examinations. Good communication skills and a systematic but flexible approach are needed to gather as much useful information as possible. There are procedures throughout the examination process that can be very helpful. For example, these include questionnaires that ensure that no important aspects of the disease and its epidemiology are omitted. The way in which interviews are conducted and the order in which they are conducted is also helpful and important. For example, talking to farmworkers first, whenever possible, helps to find out what they have to say without being influenced by the farmer’s version. The manner of speaking must be adapted, taking the interlocutor’s knowledge and language proficiency into consideration. It is necessary to listen actively to what is being said and to give feedback in one’s own words to confirm that what one has understood is correct. At this stage, the examiner should stick to the raw version of the history, which means not interpreting words and substituting them with medical terms as, e.g., “haemorrhagic lesions” instead of “bleedings in the skin.” This allows the examiner to stay open-minded and avoid missing essential information. Targeted questions can then be asked for more in-depth details, but the risk of obtaining biased or suggested answers must be considered. Furthermore, it should be taken into consideration that the owners of the farm and/or the farmworker are under stress and want to present themselves in the best light. They will (un)intentionally try to avoid giving the interviewer the impression that their actions—no matter whether illegal or not—are responsible for disease introduction. 

For a targeted collection of valid information, knowledge from communication and social science, psychology and biological/veterinary knowledge about ASF must be combined.

Epidemiological investigations, especially in backyard holdings in village settings, remain difficult. The ways of ASFV introduction and spread are not fully understood or seen; in particular, the ways of spread from backyard to backyard are still unclear, but human activities (farmers in contact, butchers and veterinarians) do play a major role [5,14]. 

**Table 1 pathogens-11-00702-t001:** Statements and opinions about ASF collected from lay persons during the outbreak investigations and grouped according to their plausibility.

Nr.	Statement	Comments	Plausibility *
1	Two days ago, we bought three piglets from a farm located in an ASF restricted area. These piglets have brought the virus into our holding and infected our pigs which died yesterday.	Incubation period is longer than one day [15].	-
2	My neighbor is a hunter. He hunted wild boar in Eastern Europe last summer. He might have brought the virus with him that infected my pigs.	Time gap too long [15].	-
3	We never feed kitchen scraps from restaurants or unknown sources, only scraps from our own kitchen.	Swill feeding is a known risk factor [16].	+
4	The window at the end of the barn was broken for over a week until we replaced it. I assume that the virus entered through the broken window.	No evidence of airborne transmission [17].	-
5	Grandfather goes to the forest every week to pick mushrooms. On Sundays he visits us for lunch. Maybe he brought the virus on his shoes.	Quite a contrived scenario, very unlikely [17].	-
6	Three weeks ago, workers came to us to repair the water supply in the stables. The same day, they were previously on a farm that tested positive for ASF.	Within high-risk period, possibly gap in biosecurity [16].	+
7	The neighboring farm which was tested ASF positive last week has done rodent control two weeks ago. After that we saw many rats. Maybe the rats, brought the virus to our farm.	Mammals and flies unlikely to be mechanical vectors [18].	?
8	Crows come from the forest and sit on the roof of our stables. They bring the virus from the forest.	Minor risk of ravens carrying meat pieces to stables [19,20].	?
9	Three months ago, most of the farm workers were replaced by new employees.	This could be a gap in biosecurity (lack of knowledge) [3].	+
10	In fact, we were able to prove that our farm was free of ASF. We tested 10% of the animals by Antibody-ELISA two days before the outbreak and all tested animals were negative.	Usually it takes more than 10 days until antibodies are detected [15].	-
11	We have repaired all the fences and renovated all the buildings. The virus cannot enter our farm.	Biosecurity includes management practices as well not only physical barriers [3,16].	-
12	In recent weeks, pigs have been dying in neighboring villages; there is nothing I can do.	Backyard scenario often seen in Central and Eastern Europe [5,14].	+
13	Wild boar come very close to the farm. They contaminate the environment.	Direct or indirect contact of wild boar with domestic pigs cannot be excluded [21].	+
14	My daughter’s boyfriend is from the farm where there was an outbreak 5 weeks ago. They have had regular contact, although I am against this relationship. Now he has brought us the virus.	Direct or indirect contact cannot be excluded in combination with insufficient biosecurity [16].	?
15	My farm is located on the main road; all animal transporters and rendering vehicles pass by here. The virus came from the tires of one such truck.	Direct or indirect contact cannot be excluded in combination with insufficient biosecurity [22].	?
16	The veterinarian came here last week and vaccinated my pigs, few days later the first animals became sick. He brought the virus.	Iatrogenic transmission cannot be excluded [22].	+
18	A few days ago, the pigs got new bedding material. It was harvested from a field not far from here.	Transmission through the bedding material cannot be excluded [23].	+
19	The week before the pigs died, it was very hot and there were a lot of blood sucking insects around. They could have brought in the virus.	Tabanids and vectors could be mechanical short-distance vectors but do not act as true vectors [22].	?
20	I found a bone in the bedding material. This could be the source of the introduction.	Transmission through the contaminated bedding material cannot be excluded [23].	+
21	My neighbour is a hunter. His dog often comes to our garden. The dog has brought the virus to us.	Vertebrates other than pigs are not known as active vectors, but they pose a minor risk for mechanical transmission. Some dogs have a strong instinct to bury their food and if they can freely run around, they can bring a piece of a wild boar to the farm and contaminate something from where or with what the virus could be carried to the pigs [20].	?
22	I visited an infected pig farm and saw a hedgehog in the pen. Maybe he brought the virus into the stable.	Hedgehogs are not known as active or passive vectors [20].	-
23	In the infected pigsty there were many stable flies. They were everywhere—flying around, sitting on the surface of the pens and on the pigs, eating the feed in the trough. The windows were not protected with nets. Maybe the flies brought the virus from the nearby forest.	Tabanids and vectors could be mechanical short-distance vectors but do not act as true vector [22].	?
24	Near each entrance to the barn are disinfecting mats for shoes. A wheelbarrow is used to clean the pens from pig manure. However, it is not possible to clean and disinfect the wheelbarrow after each use (between the barn and manure storage). Probably the virus was introduced into the barn with the wheelbarrow.	Low contagiousness, would require a high viral load on the wheelbarrow, but cannot be completely ruled out.	?
25	Due to very hot weather conditions all doors and windows of the stable were left open day and night for couple of weeks. During two days last week there were very strong winds and I think that ASF virus was brought into the stable by these winds, because pigs near the door and window were the first showing disease symptoms.	No evidence of airborne transmission by winds.	-
26	The price of pork is so cheap on the other side of the border, and I have heard that many pigs die. But people from abroad come to us to buy cheap alcohol and cigarettes.	Panic sales of pigs due to illness may lead to drop in meat price and higher purchasing activities.	+
27	Only my guard dogs, which are in front of the pigsty, get meat scraps, but not my pigs.	Dog could bring the meat (bones) to the pigs.	?
28	The manure truck removed pig manure from the storage basin recently. The manure was disposed on the nearby fields as a fertilizer.	Contamination of farmyards by vehicles, particularly through dirty tires, cross contamination possible [16].	?
29	Pigs walk outside from one pen to another, following their own path.	Direct or indirect contact with infected pigs or material (contaminated environment) Outdoor keeping requires additional biosecurity [8].	?
30	Last month, construction work was carried out in the stables, but it involved employees of a specialized company who had no contact with pigs.	Gaps in biosecurity cannot be excluded [3,16].	?
31	Only one horse is kept in the stable where the ASF-infected sow lived.	Other animals in the barn pose a biosecurity risk, e.g., if the horse goes frequently in and out.	?

*: (-) Implausible, makes no sense, can be neglected; (?) rather implausible—should not be ignored entirely, but eventually further investigated; (+) Plausible, makes sense, needs to be taken seriously.

## Data Availability

Not applicable.

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
