# Peer review of "African Swine Fever Outbreak Investigations—The Significance of Disease-Related Anecdotal Information Coming from Laypersons"

_pathogens, 2022, doi:10.3390/pathogens11060702_

Round 1

Reviewer 1 Report

In the article entitled: "African swine fever outbreak investigations – The significance of disease related anecdotal information coming from laypersons" authors provided an information on so-called “anecdotal information” and their interpretation by authorities responsible for disease control.

ASF has become an emerging problem that have enormous socio-economic impact on pork trade worldwide. Despite the disease has been introduced over 60 years ago to the Europe and was investigated by many researchers, there are still major knowledge gaps. Among them, one is related to transmission into the pig farms, especially in farms with high biosecurity level. From this point of view this report is interesting and justified.

Please find below minor remarks/comments:

11.  Please follow the citation style of Pathogens (i.e. lines 40-41) throughout the manuscript

22.   I cannot agree completely with the statement in lines 122-123. In fact Olesen et al. proved that aerosol route of infection exist. Therefore, air route should be considered as possible in the near distances[1]

33.    Line 137: “I.e. example statement No. 7.”

44.   Authors should consider discussion that gathered interview may be simple influenced by lies of farmers or staff, just to obtain compensation of losses, caused by i.e. not following biosecurity rules.

In conclusion, Authors of the communication correctly described and interpreted the problem of “in-field” interview. I recommend publishing this article. I’ve got no further comments.

1.         Olesen, A.S.; Lohse, L.; Boklund, A.; Halasa, T.; Gallardo, C.; Pejsak, Z.; Belsham, G.J.; Bruun, T.; Bøtner, A. Transmission of African swine fever virus from infected pigs by direct contact and aerosol routes. Vet. Microbiol. 2017, 211, 92–102, doi:10.1016/j.vetmic.2017.10.004.

Author Response

We would like to thank the reviewer for the constructive suggestions and corrections. Our comments on the individual points can be found below.

Rev.: Please follow the citation style of Pathogens (i.e. lines 40-41) throughout the manuscript

We will correct this accordingly together with the editorial office.

I cannot agree completely with the statement in lines 122-123. In fact Olesen et al. proved that aerosol route of infection exist. Therefore, air route should be considered as possible in the near distances[1]

Thank you for the comment and literature suggestion. We have modified the text accordingly by introducing the following sentence: “However, it has been shown that over very short distances, transmission can also take place via the air (Olesen et al., 2017)”

Rev.: Line 137: “I.e. example statement No. 7.”

We have corrected accordingly

Authors should consider discussion that gathered interview may be simple influenced by lies of farmers or staff, just to obtain compensation of losses, caused by i.e. not following biosecurity rules.

Thank you for this valid comment. We have introduced following sentences: “Nor can it be ruled out that reports will always be truthful. Reporters could be attempt to exonerate themselves from blame if certain requirements (e.g. concerning farm biosecurity) were not followed.”

Reviewer 2 Report

This study presented by Lamberga and colleagues is aimed to report statements and opinions about ASF collected from laypersons during the outbreak investigations and grouped according to their plausibility. The manuscript is tricky in several passages, abstract and discussions should be improved. In addition, extensive editing of English language and style is required. In its present form, the paper is not acceptable and requires some corrections before it could be published in animals. If you are prepared to undertake the work required, I would be pleased to reconsider my decision. Please find more specific comments below.

The affiliation of the authors must be formatted according to MDPI. Add the affiliation number to each author.

Please, format all references according to mdpi 

INTRODUCTION

Line 46 please change determine with Investigate, will be more appropriated

Line 47  please change on the likely spread with on the potential spread

MATERIALS AND METHODS

Line 70-71 during the investigations conducted in Baltic States, in which countries did you perform this survey? All? Or some of them? Please be more specific.  

Line 71 Serbia is not an EU country

Table 1 could provide more information like who said the statements and the countries. Or all these statements are reported in all the countries without differences? 

Table 1, second box 

Could this be categorised as “Statements that are rather implausible but should not be ignored completely”? Maybe the event of last summer was not correlated but similar event may occur more recently.

Table 1, fourth box

What about the possibilities that flights enter in the farm and act as mechanical vector?

Table 1, fifth box

Also in this case this statement should not completely ignored and could enter in the category Statements that are rather implausible but should not be ignored completely. It is controversial but raise awareness on the important of using different shoes for visiting forest and visit farms should be considered.

Author Response

We would like to thank the reviewer for the constructive suggestions and corrections. Our comments on the individual points can be found below.

Rev.: In addition, extensive editing of English language and style is required. In its present form, the paper is not acceptable and requires some corrections before it could be published in animals.

There must be a misunderstanding, it is not intended to publish the manuscript in "animals" but in "pathogens". Regarding the language, extensive English revisions have been performed by a native speaker who is also a world-renowned expert in the field of ASF.

Rev.: The affiliation of the authors must be formatted according to MDPI. Add the affiliation number to each author.

We will correct this accordingly together with the editorial office

Rev.: Please, format all references according to mdpi 

We will correct this accordingly together with the editorial office.

Line 46 please change determine with Investigate, will be more appropriated

We have replaced "determine" with "find out" because the suggested word "investigate" is already in the sentence and therefore it does not fit well linguistically.

Rev.: Line 47  please change on the likely spread with on the potential spread

We have changed accordingly

Rev.: Line 70-71 during the investigations conducted in Baltic States, in which countries did you perform this survey? All? Or some of them? Please be more specific.

The names of the Baltic States have been specified  

Rev.: Line 71 Serbia is not an EU country

The sentence does not refer to EU countries, but to countries in Central and Eastern Europe, to which Serbia also belongs. To avoid misunderstandings, we have added the Balkans.

Rev.: Table 1 could provide more information like who said the statements and the countries. Or all these statements are reported in all the countries without differences? 

Regarding the collected statements we were only concerned with the subjective character of such statements. Who exactly reported this in which country is completely irrelevant and would not change anything about the subjective character of the statements.

Rev.: Table 1, second box; Could this be categorised as “Statements that are rather implausible but should not be ignored completely”? Maybe the event of last summer was not correlated but similar event may occur more recently.

One attempt in our communication was to show that interpretation of statements can vary depending on the interpreter, as we said in the discussion. Therefore, investigators, perhaps less familiar or experienced with the epidemiology of ASF, might arrive at a different interpretation. Of course, events similar to those described may have occurred more recently, but that was not the case here.

Rev.: Table 1, fourth box; What about the possibilities that flights enter in the farm and act as mechanical vector?

As mentioned above interpretation of statements can vary depending on the interpreter, as we said in the discussion. Of course, the possibilities that flights enter in the farm cannot be excluded. However, their role as mechanical vectors over longer distances, coming from outside the farm and carrying the virus, are not known to us. Nevertheless, it can be speculated. However, the assessment is up to the expert conducting the investigation and depends also on the local conditions.

Rev: Table 1, fifth box; Also in this case this statement should not completely ignored and could enter in the category Statements that are rather implausible but should not be ignored completely. It is controversial but raise awareness on the important of using different shoes for visiting forest and visit farms should be considered.

Of cause, theoretically it is an indication that there were connections to the forest, but for epidemiological evidence the facts are missing. See also the comments above regarding the interpretation of statements. One should always be careful not to believe any stories or rumours without verification and thus be distracted from the real events.

Round 2

Reviewer 2 Report

The authors have improved the quality of the paper. However, this manuscript still requires some editorial corrections to the text. In light of these considerations, I recommend the publication of this manuscript with only a few minor corrections. Please find more specific comments below.

The affiliation of the authors must be formatted according to mdpi. Add the affiliation number to each  bibliographic references.

LINE 127 Please correct the typing error

Author Response

We would like to thank the reviewer for the suggestions. Our comments on the individual points can be found below.

Rev: The affiliation of the authors must be formatted according to mdpi. Add the affiliation number to each  bibliographic references.

Has been corrected accordingly.

Rev: LINE 127 Please correct the typing error

Has been corrected accordingly.
